# The Humanoid Robot Sil-Bot in a Cognitive Training Program for Community-Dwelling Elderly People with Mild Cognitive Impairment during the COVID-19 Pandemic: A Randomized Controlled Trial

**DOI:** 10.3390/ijerph18158198

**Published:** 2021-08-03

**Authors:** Eun-A Park, Ae-Ri Jung, Kyoung-A Lee

**Affiliations:** 1College of Nursing, Bucheon University, Bucheon 14774, Korea; pea0701@bc.ac.kr; 2Research Institute of Nursing Science, Seoul National University, Seoul 03080, Korea; tj720221@snu.ac.kr; 3Korea Healthcare and Policy Association, Seoul 04555, Korea

**Keywords:** mild cognitive impairment (MCI), humanoid robots, randomized controlled trial, cognition, depression

## Abstract

Background: Mild cognitive impairment (MCI) is a stage preceding dementia, and early intervention is critical. This study investigated whether multi-domain cognitive training programs, especially robot-assisted training, conducted 12 times, twice a week for 6 weeks can improve cognitive function and depression decline in community-dwelling older adults with mild cognitive impairment (MCI). Methods: A randomized controlled trial was conducted with 135 volunteers without cognitive impairment aged 60 years old or older. Participants were first randomized into two groups. One group consisted of 90 participants who would receive cognitive training and 45 who would not receive any training (NI). The cognitive training group was randomly divided into two groups, 45 who received traditional cognitive training (TCT) and 45 who received robot-assisted cognitive training (RACT). The training for both groups consisted of a daily 60 min session, twice a week for six weeks. Results: RACT participants had significantly greater post-intervention improvement in cognitive function (*t* = 4.707, *p* < 0.001), memory (*t* = −2.282, *p* = 0.007), executive function *(t* = 4.610, *p* < 0.001), and depression (*t* = −3.307, *p* = 0.004). TCT participants had greater post-intervention improvement in memory (*t* = −6.671, *p* < 0.001) and executive function (*t* = 5.393, *p* < 0.001). Conclusions: A 6-week robot-assisted, multi-domain cognitive training program can improve the efficiency of global cognitive function and depression during cognitive tasks in older adults with MCI, which is associated with improvements in memory and executive function.

## 1. Introduction

### Demographic Changes and Mild Cognitive Impairment

Mild cognitive impairment (MCI) is a stage preceding dementia that does not meet the dementia criteria but leads the affected individual to show reduced cognitive function in such domains as memory and language in comparison with healthy individuals of the same age group and with similar levels of education; MCI individuals may show normal or slightly reduced daily activities [1]. It is estimated that about 12% to 36% of older adults have MCI, and as the population of older adults increases, the prevalence of MCI will gradually increase [2,3]. The number of MCI elders in South Korea in 2019 showed a 19-fold increase compared with 2009, accounting for 22.7% of the total elderly population; the reported number was approximately 276,045 people [4,5].

Approximately 1–2% of older adults with normal cognitive functions but approximately 10–15% of those with MCI would show the development of Alzheimer’s dementia [6]. Because MCI is a clinical stage in the determination of pathological changes toward dementia, early intervention is critical [7].

Subjective memory complaint (SMC) in healthy older adults is associated with depression, which has been related to cognitive impairment, such as dementia, and it is known that SMC is prominent in healthy older adults living in the community [8]. Therefore, identifying SMC elders is expected to allow the prevention of cognitive impairment of those in the healthy senile stage. To compare SMC with MCI, SMC elders show no objective memory impairment (OMI), while MCI elders are characterized by OMI in early diagnosis [9] with a 2-fold higher risk of dementia compared with healthy older adults [10].

The early treatment of MCI, which may be viewed as an early stage of dementia, as soon after detection as possible, is likely to substantially delay the onset of dementia and functional deterioration. Thus, the early screening of MCI elders to protect their cognitive function and prevent depression, as well as to maintain the activities of daily living (ADL) at a healthy state, is crucial in the prevention and management of dementia [11].

In the treatment of MCI elders, drug interventions are not effective for long-term improvement of cognitive function, with downsides such as drug side effects or accelerated progression of dementia symptoms [12]. This has placed an emphasis on drug-free interventions, and previous studies have attempted various cognitive training interventions for cognitive function improvement.

According to a previous study analyzing past interventions [13], multi-domain cognitive training programs are more appropriate for increasing neuroplasticity. In most studies, a single mode of therapy, such as occupational therapy, kinesiotherapy, or art therapy, was more frequently applied [5], and the cognitive training only targeted a single cognitive domain, such as memory [14,15]. A single-domain cognitive intervention may have theoretical importance since it may allow researchers to investigate direct training-related effects [16], but multi-domain training could potentially have more practical advantages because multiple cognitive functions are required for humans to live in the world [17]. It is therefore possible that multi-domain cognitive training would elicit more synergistic transfer effects across domains than single-domain cognitive training since it targets multiple cognitive functions [18,19].

Computerized cognitive training programs have increasingly replaced tasks that were originally paper-and-pencil format with computer-based tasks for practice and training [20], but traditional human-assisted cognitive training (TCT) with the former usually needs experienced instructors [21], and those qualified instructors may be unavailable in some chronic care facilities or community centers. In particular, during the COVID-19 pandemic, people with dementia are particularly vulnerable to COVID-19 due to their age, multimorbidity, and difficulties in maintaining physical distancing [22,23]. As the illness and consequent distancing may increase family carer stress, loneliness, neuropsychiatric symptoms, and the use of psychotropic medication—leading to complications, including future dementia—there is an urgent need for technological interventions based on robot-assisted cognitive programs [24].

Recently, studies have reported on the use of an advanced robot to delay cognitive impairment and reduce depressive symptoms in older healthy adults through a 12-week, 5-days per week, 90 min per session intervention involving 17 robot-assisted cognitive training (RACT) programs. When the robot and traditional interventions were directly compared, the robot group showed less cortical thinning in the anterior cingulate cortices [25]. In the study of Pino et al. [26], the rate of cognitive impairment could be delayed, and depressive symptoms could be reduced in MCI elders through interaction and memory training. Manca et al. [27] presented the impact of 12 sessions a month of serial music games using humanoid robots and tablets with 14 elders with MCI, measuring their response and level of improvement. However, one limitation of the study is that the indicators measuring the cognitive status improvement were too simple and the number of participants was too small. ICT-based cognitive training programs have so far exhibited inconsistent effects in MCI elders [28], and there is a general lack of studies investigating the effects of RACT programs on MCI elders. Thus, this study tested the hypothesis that robot-assisted, multi-domain cognitive training would result in greater effects than traditional cognitive training (TCT) since exposure to new technology is likely to be more challenging for the elderly than familiar technologies and since novelty may increase global cognitive function and reduce depression. To test these hypotheses, we compared cognitive, memory, and executive functions and depression in an RACT group with a TCT group and compared the RACT group with an NI group that did not have cognitive training.

## 2. Materials and Methods

### 2.1. Design

This randomized controlled study on the effects of 6 weeks of cognitive training on improvement of cognitive function and depression in SMC and MCI participants was conducted between 28 December 2020 and 26 February 2021 at S-City Dementia Center in Gyeonggi-do, Republic of Korea.

### 2.2. Participants

Participants were enrolled according to the following inclusion criteria: (1) older adults aged 60 years or above, who reside in S-City, Gyeonggi-do; (2) individuals who are willing to participate in the study with fluent communication ability; (3) individuals who are “normal” in the MMSE-DS and who checked at least one item in the SMCQ; (4) individuals who show “cognitive decline” in the MMSE-DS but are determined to be “Normal” or “MCI” by a specialist based on CERAD-K and SGDS-K results.

Exclusion criteria were (1) a problem in ADL and instrumental ADL (IADL); (2) a diagnosis of major neurocognitive disorder (defined using DSM-5 criteria); (3) history of symptomatic stroke; (4) history of other central nervous system diseases; and (5) serious medical or psychiatric illness that would interfere with study participation.

Informed consent was obtained from all the patients or from their legal representatives when appropriate.

In terms of the required sample size, a total of 120 participants was derived using the G*Power 3.1.2 program [29]. This sample size was determined for a repeated measures ANOVA with three independent groups, using a significance level of 0.05, a statistical power of 0.80, an effect size of 0.25 (which indicates a moderate effect for an ANOVA), and two repetitive measurements. The desired statistical power must amount to approximately 10% to 20% more than the required sample size [30], and we considered the COVID-19 pandemic, the characteristics of variable elderly health status, and visits to the dementia center for 6 weeks. Therefore, we considered a 30% dropout rate in the number of recruited sample, so a maximum of 156 participants had to be recruited. In fact, 175 people were finally recruited.

Among the 175 individuals recruited during a period of two weeks, 15 were excluded based on the exclusion criteria, and the remaining 160 individuals were randomly divided into three groups, RACT (54 participants), TCT (53 participants), and NI (53 participants), after matching based on gender, age, and years of education. The intervention was performed, and after excluding six individuals who dropped out of the study because of health reasons (RACT 4 and TCT 2) and 19 individuals who refused to take the post-test due to the COVID-19 pandemic (RACT 5, TCT 6, and NI 8), the number of participants in the final analysis was 45 in RACT, 45 in TCT, and 45 in NI (Figure 1).

### 2.3. Materials

#### 2.3.1. Main Software

RACT was implemented with older adults aged 60 years or above at three dementia centers and six elderly welfare centers, as part of the efforts taken by S-City, Gyeonggi-do, to care for the socially vulnerable, reduce carer burden, delay the rate of dementia, and improve cognitive function in the elderly. The program was carried out using the Sil-bot, with intervention content developed for the purpose of dementia prevention and delay by providing cognitive training toward the activation of brain functions in participants [26].

The humanoid robot, Sil-bot (Robocare, Seongnam, Korea) was used in the RACT group (Figure 2 and Figure 3). Participants in the robot group responded to instructions using an individual smart pad, except for three programs for which the robots detected and evaluated the participants’ motion (Galaxy tab 10.1, Samsung Electronics, Seoul, Korea). Instructors taught the participants in the procedures used for cognitive training with the robot. The main roles of instructors in the robot group were to select programs for participants as well as to give or repeat program instructions if necessary. Unlike the TCT, RACT rewarded participants by giving an individual feedback to each question immediately after entering the individual submitted an answer on their smart pads. Individual scores were saved at this point. A total of 22 cognitive training programs were used in the RACT, including 6 programs for memory, 3 for concentration, 2 for memory and concentration, 1 for concentration and visual identification, 2 for reasoning and judgement, 1 for visual comprehension, 1 for language, 1 for calculation, 1 for calculation and concentration, 2 for visuospatial function, and 2 for physical activity. Sil-bot is able to display various facial expressions and emotions based on humanoid–robot interaction (HRI) technology, while it is also capable of moving its arms, shoulders, and head and capable of performing whole direction movements using its bottom wheel.

As a humanoid robot, Sil-bot can recognize user intent, gaze, and emotional expressions, while being familiar to the users. It can also move based on an autonomous driving system. Despite being a robot loaded with advanced technology, Sil-bot can be used by non-professionals, so it was applied as an assistant instructor in the cognition-enhancing program performed in this study (Table 1).

#### 2.3.2. Variables and Measuring Instruments

##### Cognition

Participants’ cognition was measured by 19 items in the Mini-Mental State Examination–Dementia Screening (MMSE-DS) developed by Folstein et al. [31] and standardized to meet Korean needs by Kim et al. [32]. The MMSE-DS has seven subdomains, which are time orientation (5 items), place orientation (5 items), attention (1 item), memory (2 items), speech ability (3 items), construction ability (1 item), and judgment ability (2 items). The range of total scores is 0 to 30 points, and the higher the score, the better the cognition. The cutoff point varies depending on gender, age, and years of education. For reliability, Cronbach’s α = 0.83 in Kim et al. [13], and in this study, Cronbach’s α = 0.82.

##### Subjective Memory Complaints

Developed by Youn et al. [33], the Subjective Memory Complaint Questionnaire (SMCQ) is a tool comprising questions on the subjective mood and experience of memory impairment. Each of the 14 questions are based on a 2-point Likert scale, “Yes” and “No”, with the score range of 14–28. Higher SMC scores indicate higher reduction in memory. The tool reliability at the time of development was Cronbach’s α = 0.86, while in this study, the reliability was Cronbach’s α = 0.84.

##### Neuropsychological Assessment

Neuropsychological assessment is a test developed by the Consortium to Establish a Registry for Alzheimer’s Disease (CERAD), applied to the screening of senile cognitive decline or MCI. The tool validity and reliability were verified in Lee et al. [34], after which the translated and standardized form of the test, the Korean version of CERAD (CERAD-K), has been in use. In this study, the CERAD-K included eight tests: (1) Verbal Fluency: language generation, semantic memory, executive function, (2) Modified Boston Naming: naming ability, (3) MMSE-KC: orientation, language, concentration, memory construction, (4) Word List Memory: verbal memory, learning ability, (5) Constructional Praxis: constructional praxis, (6) Word List Recall: delayed recall, (7) Word List Recognition: recognition memory, (8) Constructional Recall: visuospatial memory. The administration time for the test is approximately 40 min, and the tool reliability at the time of development was Cronbach’s α = 0.92, and in this study, Cronbach’s α = 0.94.

##### Depression

The Geriatric Depression Scale Short Form: Korean Version (GDSSF-K), a tool developed by Yesavage and Sheikh [35] and translated and modified by Kee [36], was used to measure depression. The GDSSF-K consists of 15 items, each assigned a score of 1, with a total score range of 0–15. Higher scores indicate higher severity of depression. The tool reliability was Cronbach’s α = 0.72 in Kee [36], and Cronbach’s α = 0.92 in this study.

#### 2.3.3. Data Analysis

For data analysis, IBM SPSS 22.0 (IBM Corp., Armonk, NY, USA) was used. Descriptive statistics were calculated to explore the frequency, percentage, mean, and standard deviation of all the variables. For homogeneity analysis of the three groups, ANOVA and X^2^ tests were used. The normality of each variable was tested using the Kolmogorov–Smirnov test. The differences between pre- and post-test scores of cognitive function and depression for the three groups were analyzed using the paired *t*-test, while the differences among the RACT, TCT, and NI groups were analyzed via ANOVA, with Tukey’s test in post hoc analysis. The differences in test scores depending on time and groups were analyzed using repeated measures ANOVA. The level of significance was set to *p* < 0.05.

### 2.4. Procedure

After receiving a 16 h training from a specialist, the RACT program was performed by the nurses, occupational therapists, and social workers at the dementia centers and elderly welfare centers. The trained instructors first held an orientation for the participants on the purpose and content of the study, and considering that the participants were older adults, safety and disinfection were the top priority. The program instructors also recorded progress in daily reports, while sharing the current status with the investigators in real-time through SNS. Data collection was performed by the instructors at the dementia centers, who were trained by the National Dementia Institution (NDI) on the use of the measurement tools of MMSE-DS, SMCQ, CERAD-K, and GDSSF-K. The principal investigator did not participate in the data collection process. Data collection was performed under double-blind conditions to prevent the raters and participants from being able to identify those in the control and those in the intervention groups. In the case of an illiterate participant, the rater read each question aloud slowly, recorded the participant’s response, and verbally checked if it was correct.

#### 2.4.1. RACT Groups

The RACT groups were eight teams of 45 participants, and each team was given 12 sessions of RACT, twice a week for six weeks in total. Among the 22 cognitive training programs, the instructor selected 3 to 4 programs in each session. The participants were guided to solve problems on smart pads or performed a physical, musical, or art activity or received sensory stimulation therapy, while the robot displayed around 30 emotional expressions to compliment and encouraged the participants through the software linked to the smart pads that relayed the points. The instructors completed a ≥8 h program training by the professional instructor prior to the study.

#### 2.4.2. TCT Groups

The TCT groups were six teams of 45 participants, and each team was given 12 sessions of TCT; the “Dugeun Dugeun Brain Fitness” program, which was originally in paper-and-pencil format, was given twice a week for six weeks in total. The aforementioned program was developed by the NDI based on 24 different brain fitness activities, and the instructor selected 3 to 4 activities per session for the training. The program was constructed to train the participants in such cognitive functions as memory, orientation, judgment, concentration, restraint, computation, and visuospatial and language abilities, all of which are easily affected by aging or dementia. The instructors completed a ≥24 h program training by the NDI prior to the study, who has implemented the program for several years.

#### 2.4.3. NI Groups

The NI (control) group was given no intervention, although they received the pre-test and the post-test six weeks later. After the completion of the study, the participants in the NI group were provided with the program if they wished.

### 2.5. Ethical Considerations

For ethical consideration, data collection began after approval by the IRB (KoNIBP: P01-202012-11-001), and this study is registered in CRIS (clinical research information service), one of the WHO’s international clinical trials registry platforms (registry number: KCT0006377).

Before distributing the questionnaire, the authors explained the following. Collected data would be used for research purposes only. All specific gains and losses from participation in the study were presented. After explaining the study, if the participants agreed to take part, issues of anonymity, confidentiality, and freedom to withdraw from the study and that all the data would be discarded after study completion were described, after which the research participation agreement was prepared. A total of two copies of the written consent were created; one was given to the participant, and the other was stored separately from the questionnaire. All collected data, including questionnaires and computational files, will be discarded three years after the end of the study.

## 3. Results

### 3.1. Sample Description

A total of 135 participants were enrolled in this study: 37 males (27.4%) and 98 females (72.6%), with a mean age of 75.9 ± 6.1 years. The mean length of education was 8.8 years, and the final level of education of the largest number of participants was elementary school (43 participants, 31.9%). The cognitive impairment status was 57 MCI (42.2%) and 78 SMC (57.8%). Before intervention, the age, gender, living status, number of children, years of education, health status, cognitive function, and comorbidity status of the three groups showed no significant difference across all variables in the test of homogeneity (Table 2).

### 3.2. Effect of Humanoid Robot Sil-Bot in a Cognitive Training Program on Cognition and Depression

The scores of cognitive function and depression based on the intervention are presented in Table 3. Before the intervention, the three groups exhibited uniform levels of cognitive function and depression. After intervention, the RACT group showed the lowest scores of SMCQ and GDSSF-K (4.7 ± 3.5 and 3.0 ± 3.6), with a significant difference (*p* < 0.05). The pre- and post-test results of the three groups were analyzed using repeated measures ANOVA, where significant time and group interactions were found for both cognitive function and depression. Hence, the changes in cognitive function (*F* = 6.172, *p* = 0.003), SMC (*F* = 14.635 *p* < 0.001), depression (*F* = 6.284, *p* = 0.002), and neuropsychological assessment (*F* = 5.274, *p* = 0.006) between pre- and post-intervention were shown to be significant for RACT, TCT, and NI groups (Table 3).

### 3.3. Differences in the Pre- and Post-Intervention Effects

The before and after differences based on the intervention are presented in Table 4.

#### 3.3.1. Cognition

The MMSE-DS scores in the RACT groups increased from pre-test 25.3 ± 4.1 to post-test 26.6 ± 3.3 (*t* = 4.707, *p* < 0.001), while the scores in the TCT and NI groups showed no significant change (Table 4).

#### 3.3.2. Subjective Memory Complaints

The SMCQ scores showed an increase from pre-test 5.9 ± 3.3 to post-test 4.7 ± 3.5 in the RACT groups (*t* = −2.282, *p* = 0.007) and from pre-test 7.6 ± 2.0 to post-test and 5.0 ± 3.2 (*t* = −6.671, *p* < 0.001) in the TCT groups, indicating a statistically significant reduction, while the NI groups showed no significant change before and after intervention (Table 4).

#### 3.3.3. Neuropsychological Assessment

The CERAD-K scores for cognitive functions showed a significant increase in the RACT and TCT groups (*p* < 0.001), while the NI groups showed no significant change before and after intervention. For each item of the test, a significant change was found in Boston Naming, MMSE-KC, Word List Memory, Word List Recall, and Word List Recognition for the RACT groups and in Word List Memory, Word List Recall, and Constructional Recall for the TCT groups (Table 4).

#### 3.3.4. Depression

The GDSSF-K scores for depression in the RACT groups showed a significant decrease from pre-test 4.3 ± 4.8 to post-test 3.0 ± 3.6 (*t* = −3.307, *p* = 0.004). The TCT and NI groups showed no significant change before and after intervention (Table 4).

### 3.4. Differences in the Pre- and Post-Intervention Effects According to General Characteristics

The differences in pre-post test scores according to gender, age, and years of education were analyzed. In the RACT groups, there were statistically significant differences in pre–post score of MMSE-DS, SMCQ, GDSSF-K, and CERAD-K according to gender, age, and years of education (*p* < 0.05). In particular, it was analyzed that the results were more consistent when participants were female depending on gender, 75 years or older depending on age, 9 years of education or less depending on the education level (*p* < 0.05).

In the TCT groups, there were statistically significant differences according to gender, age, and years of education in the SMCQ score and CERAD-K score (*p* < 0.05), and the GDSSF-K score showed a statistically significant difference only in the group over 75 years old (*t* = −2.522, *p* = 0.018). In the NI group, there was no statistically significant difference according to the general characteristics of the participants (Table 5).

## 4. Discussion

This study was conducted to determine the effects of an RACT program on improving cognitive function and depression in SMC and MCI elders aged 60 years or above. The findings are of considerable significance in that the effects of the RACT program on the cognitive functions and depression in MCI elders were evaluated through comparisons involving three groups, between RACT and NI as well as between RACT and TCT.

A significant difference was found in the changes in cognitive function and depression between the two experimental groups. In contrast with the NI group, the participants in the RACT and TCT groups showed an improvement in cognitive functions, while the effect of the program on depression was greater in RACT than TCT after the intervention.

Analyzing the effect of the RACT on cognitive function improvement showed that the MMSE-DS, SMCQ, and CERAD-K uniformly indicated an improvement, which agreed with previous studies in which robot-assisted, ICT-based cognitive training programs were found effective in reinforcing the different domains of cognitive function [26,37,38].

The RACT program in this study was found to have a positive effect on improving such functions as language production and memory, short-term memory, and attention, based on the results of Boston Naming, MMSE-KC, Word List Memory, Word List Recall, and Word List Recognition as part of the neuropsychological assessment. The TCT program was also found effective in cognitive function improvement, with a greater effect on subjective memory complaints and visuospatial and compositional abilities. The RACT program was shown to stimulate the cerebral cortex to improve the main cognitive functions as well as to improve memory and problem-solving abilities, while changing the cortical width of the frontal and temporal lobes as physiological indicators [25].

Among the numerous cognition-enhancing programs investigated so far, the multi-domain cognitive trainings showed more positive post-intervention effects than the those focused solely on memory [39]. This is in line with the theory that an environmental stimulus or exercise such as a cognitive training activates brain functions and increases neurogenesis and synaptic plasticity [40]. RACT consists of activities designed to enhance multiple cognitive domains: memory, attention, computation, and visuospatial ability, such as imitating the robot’s motion in a given order, walking on a square board after memorizing the path of the robot’s motion, grabbing a goody bag falling on the monitor screen, performing mental arithmetic for a money problem set by the robot, and making a square from variously shaped figures using smart pads. The changes in the robot’s motion and facial expression increased the eye gaze of MCI elders during training to improve their concentration and immersion [26,41], while the interactions with the robot allowed them to experience a complex set of physical, cognitive, and psychological activities with consequent reinforcement of cognitive function. For individuals with MCI, the ICT-based cognitive training was effective in reinforcing cognition, memory, working memory, and attention [42], highlighting the importance of training in the early stage of cognitive impairment for memory retention or enhancement.

With the advent of robotics, service robots that can interact with humans have attracted both industry and academic interest [43]. In particular, robots to assist the elderly may be important given the rapid increase in the aging population and the exorbitant healthcare costs associated with caring for older individuals with cognitive decline. Furthermore, TCT with paper-and-pencil usually needs experienced instructors [21], but such qualified instructors may be unavailable in some chronic care facilities or community centers. For this reason, we developed a total of 20 RACT programs for the elderly. In addition, the RACT program entails far less of a training burden than the TCT program, as it takes the role of an assistant rather than an instructor.

In this study, the GDSSF-K was analyzed to determine the effect of the cognition-enhancing program on depression in MCI elders. While the TCT program did not show a significant difference in depression, the Sil-bot-based RACT program was found to have considerable benefits in alleviating depression. This may be attributed to the TCT having no effect on emotional aspects such as depression, as it comprises a single passive task. Depression in older adults is not a simple psychological problem but one that is associated with reduced cognitive function, which thus necessitates a combined rather than single intervention in training to have an effect on depression and cognitive function [44]. The effect of the RACT program on depression coincided with previous studies of humanoid robots in training that had an effect in reducing depression [26,45,46]. A study on the intention to use care-robot technology revealed that the older adults were influenced by a factor of perceived enjoyment, whereas others focused more on the usefulness, adaptability, and ease of use of the robot [47]. The RACT program, compared with the TCT program, consists of a variety of contents ranging from singing favorite songs to catching money falling from the sky that could stimulate interest and joy in participants. During the study period, the COVID-19 pandemic has rapidly reduced social activities in older adults, who have consequently experienced an increase in depression caused by reduced social interactions [48,49]. As can be seen, the RACT program uses multiple interventions during training that stimulate multiple cognitive domains, thereby increasing the level of participation through increased interest and motivation while decreasing the level of depression.

According to a previous study, the prevention of a 2-point decline in the MMSE score would save about USD 3700$ annually, and a 2-point increase rather than a decline in an MMSE score would save about USD 7100$ [49]. It is far more challenging to protect cognitive abilities in older adults in the early stage of cognitive impairment than in older adults who have not yet shown a symptom of reduced cognitive function. Thus, the application of the RACT program with MCI elders to improve cognitive function or to merely delay cognitive decline is likely to have a significant effect on reducing medical costs and carer burden.

Furthermore, the motivation of the participant is critical for maximizing the effects of a successful rehabilitation program, as an elevated motivation results in far greater effects on cognitive and functional improvements [50]. The application of a novel technology of Sil-bot in a cognitive training program is anticipated to reduce depression and ensure enjoyment for participants, while the training effects may be enhanced through increased immersion and motivation [51].

The limitation of this study was that only the short-term effects were measured in a period of 6 weeks during the COVID-19 pandemic; the short-term training entailed reduced frequency and duration of the program. Thus, a mid-term to long-term follow-up study is suggested to consider the effect of the program on preventing dementia, where the program is periodically provided with an increased number of weekly sessions.

## 5. Conclusions

A 6-week robot-assisted, multi-domain cognitive training program can improve the efficiency of global cognitive function and depression during cognitive tasks in SMC and MCI elders aged 60 years or above, which is associated with improvements in memory and executive function. Therefore, it is necessary to expand the use of the RACT program as an integrated approach for improving the physical and emotional functions of the elderly, and to provide the program continuously within the facilities in which they live.

In this study, it was found that general characteristics such as gender, age, and years of education affect the effectiveness of training program. This is likely due to the fact that the use of robots and smart pads is not familiar to older people. Therefore, it will be helpful to help the elderly gradually adapt to such education through its combination with traditional training.

The RACT program could have the potential to improve robot technology acceptance, an interesting approach to bridge the digital device divide that is present among elderly people. Further studies should conduct research and development of a personalized program focused on the level of a participant’s cognitive function and study outcomes over the long term.

## Figures and Tables

**Figure 1 ijerph-18-08198-f001:**
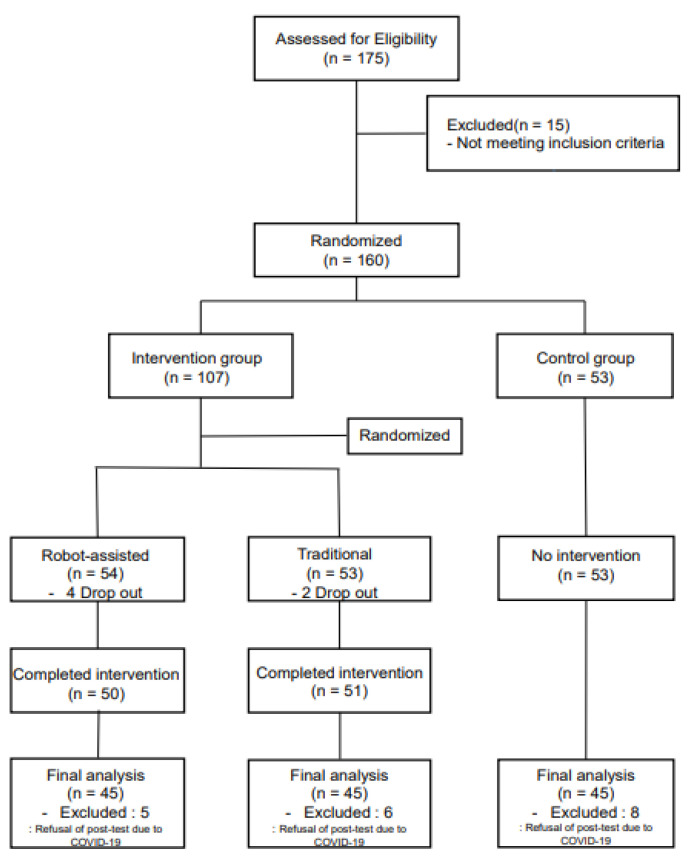
Flowchart of study participants.

**Figure 2 ijerph-18-08198-f002:**
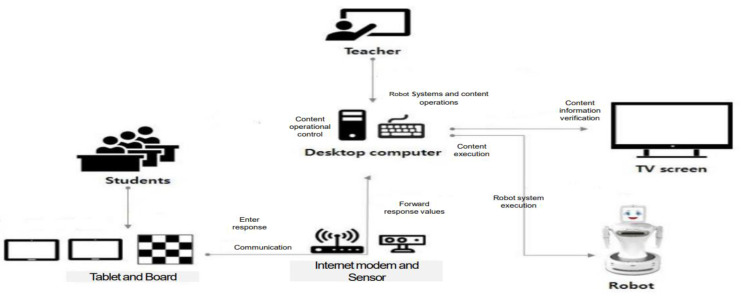
Robot system configuration chart.

**Figure 3 ijerph-18-08198-f003:**
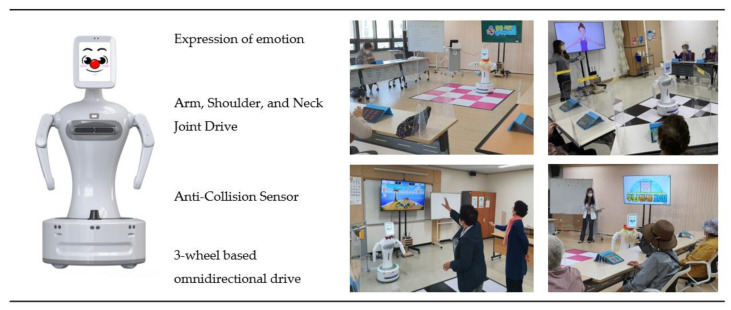
Silbot and program operation.

**Table 1 ijerph-18-08198-t001:** Composition of the Humanoid Robot Sil-bot.

Category	Detailed Specifics	Category	Detailed Specifics
Size	480 × 520 × 1150 (mm)	System	Intel i3 CPU
			8G DDR4 RAM
			2 × USB 3.0
			Wireless Ethernet
			Gigabit Ethernet
Weight	25 kg	OS	Linux Ubuntu
Battery usage time	4–6 h	Display	9.7 inch IPS decompression formula
			Display (1024 × 768 Resolution)
Battery charging time	90 min (220 Vcharger)	Sensor, etc.	Gyro sensor, LED
Battery	24 V Lithium ion	Camera	HD 720 P (1280 × 720) (mm)
Degree of freedom	11 (arms, head, mobile)	Software	ROS Kinetic
Voice and language	Korean (adult female 3 types, adult male 3 types, child 3 types)	Smart pad	Galaxytab10.1 (Samsung Electronics, Seoul, Korea)
	English (adult female 3 types, adult male 2 types)		

**Table 2 ijerph-18-08198-t002:** Homogeneity Test of General Characteristics (*N* = 135).

Variables	Categories	Total (*N* = 135)	Robot-Assisted (*N* = 45)	Traditional (*N* = 45)	No Intervention (*N* = 45)	*p*
		*n* (%)	*n* (%)	*n* (%)	*n* (%)	
Cognitive impairment status	MCI	57 (42.2)	17 (37.8)	22 (48.9)	18 (40.0)	0.529
	SMC	78 (57.8)	28 (62.2)	23 (51.1)	27 (60.0)	
Age (years), Mean ± SD		75.9 ± 6.1	75.5 ± 5.9	76.7 ± 5.9	75.6 ± 6.6	0.959
Gender	Male	37 (27.4)	13 (28.9)	12 (26.7)	12 (26.7)	0.963
	Female	98 (72.6)	32 (71.1)	33 (73.3)	33 (73.3)	
Living status (duplicate selection)	Alone	45 (33.3)	21 (46.7)	13 (28.9)	11 (24.4)	0.061
	Spouse	66 (48.9)	18 (40.0)	25 (55.6)	23 (51.1)	0.315
	Children	30 (22.2)	10 (22.2)	8 (17.8)	12 (26.7)	0.598
	Offspring	8 (5.9)	2 (4.4)	2 (4.4)	4 (8.9)	0.588
	Etc.	2 (1.5)	0 (0.0)	0 (0.0)	2 (4.4)	0.131
Number of children	None	6 (4.4)	5 (11.1)	1 (2.2)	0 (0.0)	0.189
	1	19 (14.1)	8 (17.8)	6 (13.3)	5 (11.1)	
	2	39 (28.9)	11 (24.4)	14 (31.1)	14 (31.1)	
	≥3	71 (52.6)	21 (46.7)	24 (53.3)	26 (57.8)	
Level of education	Mean ± SD	8.8 ± 4.3	9.3 ± 4.2	8.4 ± 4.6	8.8 ± 4.2	0.645
	None	12 (8.9)	3 (6.7)	5 (11.1)	4 (8.9)	0.749
	Elementary school	43 (31.9)	14 (31.1)	17 (37.8)	12 (26.7)	
	Middle school	32 (23.7)	9 (20.0)	9 (20.0)	14 (31.1)	
	High school	30 (22.2)	13 (28.9)	7 (15.6)	10 (22.2)	
	College and higher	18 (13.3)	6 (13.3)	7 (15.6)	5 (11.1)	
Health status	Average	3.1 ± 0.9	3.2 ± 0.9	3.0 ± 0.9	3.2 ± 0.9	0.906
	Very good	2 (1.5)	1 (2.2)	0 (0.0)	1 (2.2)	0.899
	good	30 (22.2)	10 (22.2)	12 (26.7)	8 (17.8)	
	Moderate	61 (45.2)	18 (40.0)	22 (48.9)	21 (46.7)	
	Bad	33 (24.4)	13 (28.9)	8 (17.8)	12 (26.7)	
	Very bad	9 (6.7)	3 (6.7)	3 (6.7)	3 (6.7)	
Comorbidity status (duplicate selection)	None	17 (12.6)	3 (6.7)	5 (11.1)	9 (20.0)	0.152
	Hypertension	70 (51.9)	27 (60.0)	25 (55.6)	18 (40.0)	0.137
	Diabetes	32 (23.7)	12 (26.7)	8 (17.8)	12 (26.7)	0.519
	Stroke	9 (6.7)	4 (8.9)	2 (4.4)	3 (6.7)	0.700
	Arthritis	42 (31.1)	17 (37.8)	9 (20.0)	16 (35.6)	0.139
	Incontinence	15 (11.1)	5 (11.1)	4 (8.9)	6 (13.3)	0.799
	Cancer	7 (5.2)	4 (8.9)	0 (0.0)	3 (6.7)	0.141
	Heart disease	17 (12.6)	6 (13.3)	4 (8.9)	7 (15.6)	0.624
	Hyperlipidemia	20 (14.8)	3 (6.7)	9 (20.0)	8 (17.8)	0.162
	Etc.	41 (30.4)	17 (37.8)	12 (26.7)	12 (26.7)	0.417

MCI, mild cognitive impairment; SMC, subjective memory complaint.

**Table 3 ijerph-18-08198-t003:** Effect of humanoid robot Sil-bot in a cognitive training program on cognitive function and depression.

Variables		Robot-Assisted ^a^ (*N* = 45)	Traditional ^b^ (*N* = 45)	No Intervention ^c^ (*N* = 45)	*p*	Source	*F*	*p*
		Mean ± SD						
MMSE-DS	Pre	25.3 ± 4.1	26.6 ± 3.0	25.8 ± 3.9	0.516	Time	3.939	0.049
	Post	26.6 ± 3.3	26.3 ± 3.1	26.0 ± 4.1	0.387	Time × group	6.172	0.003
						Group	0.335	0.716
SMCQ	Pre	5.9 ± 3.3	7.6 ± 2.0	6.3 ± 2.4	0.46	Time	31.744	<0.001
	Post	4.7 ± 3.5	5.0 ± 3.2	6.6 ± 3.0	0.008	Time × group	14.635	<0.001
					a < c	Group	2.328	0.102
CERAD-K	Pre	67.8 ± 15.1	64.5 ± 16.0	64.9 ± 13.7	0.368	Time	46.558	<0.001
	Post	71.6 ± 14.6	69.0 ± 17.4	66.0 ± 15.2	0.091	Time × group	5.274	0.006
						Group	0.925	0.399
GDSSF-K	Pre	4.3 ± 4.8	3.8 ± 3.8	4.9 ± 3.9	0.481	Time	0.949	0.332
	Post	3.0 ± 3.6	4.7 ± 4.9	4.7 ± 4.1	0.048	Time × group	6.284	0.002
					a < bc	Group	1.045	0.355

MMSE-DS, Mini-Mental State Examination-Dementia Screening; SMCQ, Subjective Memory Complaint Questionnaire; CERAD-K, the Korean version of the Consortium to Establish a Registry for Alzheimer’s Disease; GDSSF-K, the Geriatric Depression Scale Short Form: Korean Version. ^a^ RACT group. ^b^ TCT group. ^c^ NI group.

**Table 4 ijerph-18-08198-t004:** Differences in the Pre- and Post-intervention Effects.

Variables		Robot-Assisted (*N* = 45)			Traditional (*N* = 45)			No Intervention (*N* = 45)		
		Pre	Post	*t* (*p*)	Pre	Post	*t* (*p*)	Pre	Post	*t* (*p*)
MMSE-DS		25.3 ± 4.1	26.6 ± 3.3	4.707 (<0.001)	26.6 ± 3.0	26.3 ± 3.1	−1.180 (0.224)	25.8 ± 3.9	26.0 ± 4.1	0.355 (0.724)
SMCQ		5.9 ± 3.3	4.7 ± 3.5	−2.282 (0.007)	7.6 ± 2.0	5.0 ± 3.2	−6.671 (<0.001)	6.3 ± 2.4	6.6 ± 3.0	0.842 (0.404)
GDSSF-K		4.3 ± 4.8	3.0 ± 3.6	−3.307 (0.004)	3.8 ± 3.8	4.7 ± 4.9	1.971 (0.055)	4.9 ± 3.9	4.7 ± 4.1	−0.450 (0.655)
CERAD-K	Total	67.8 ± 15.1	71.6 ± 14.6	4.610 (<0.001)	64.5 ± 16.0	69.0 ± 17.4	5.393 (<0.001)	64.9 ± 13.7	66.0 ± 15.2	1.487 (0.144)
	Verbal Fluency	13.4 ± 4.2	14.0 ± 5.0	1.210 (0.233)	12.4 ± 3.9	12.9 ± 5.0	1.068 (0.292)	11.7 ± 3.7	11.7 ± 3.9	0.000 (0.999)
	Boston Naming	10.9 ± 3.2	11.4 ± 2.9	2.395 (0.021)	10.2 ± 2.8	10.5 ± 2.9	1.108 (0.274)	10.6 ± 2.6	10.6 ± 2.8	0.0119 (0.906)
	MMSE-KC	25.9 ± 3.6	26.6 ± 3.3	2.378 (0.022)	26.6 ± 3.0	26.3 ± 3.1	−1.074 (0.289)	25.8 ± 3.6	26.0 ± 4.1	0.555 (0.582)
	Word List Memory	16.0 ± 4.2	17.1 ± 4.3	2.711 (0.010)	15.7 ± 5.0	17.2 ± 5.6	2.955 (0.005)	15.5 ± 3.7	16.4 ± 4.4	1.932 (0.060)
	Constructional Praxis	9.8 ± 1.7	9.6 ± 1.9	−0.831 (0.410)	9.7 ± 1.7	9.6 ± 1.7	−0.304 (0.763)	9.4 ± 2.0	8.9 ± 2.0	−2.296 (0.027)
	Word List Recall	4.7 ± 2.3	5.4 ± 2.0	3.387 (0.001)	4.5 ± 2.7	5.3 ± 2.7	4.038 (<0.001)	4.6 ± 2.2	5.0 ± 2.7	1.633 (0.110)
	Word List Recognition	8.2 ± 2.2	8.7 ± 2.0	2.383 (0.002)	7.6 ± 2.8	7.8 ± 2.4	0.946 (0.350)	8.2 ± 2.1	8.4 ± 2.2	0.759 (0.452)
	Constructional Recall	4.7 ± 3.4	5.4 ± 3.1	1.907 (0.063)	4.7 ± 3.2	5.7 ± 3.4	3.100 (0.003)	4.9 ± 2.8	5.0 ± 2.8	0.232 (0.817)

MMSE-DS, Mini-Mental State Examination-Dementia Screening; SMCQ, Subjective Memory Complaint Questionnaire; GDSSF-K, the Geriatric Depression Scale Short Form: Korean Version; CERAD-K, the Korean version of the Consortium to Establish a Registry for Alzheimer’s Disease; MMSE-KC, Mini-Mental State Examination in the Korean version of the CERAD Assessment packet.

**Table 5 ijerph-18-08198-t005:** Differences in the Pre- and Post-intervention Effects according to General Characteristics.

Variables			Robot-Assisted				Traditional				No Intervention			
			*N*	Pre	Post	t (*p*)	*N*	Pre	Post	*t* (*p*)	*N*	Pre	Post	*t* (*p*)
MMSE-DS	Gender	Male	13	25.6 ± 3.7	26.6 ± 2.8	−1.927 (0.078)	12	26.2 ± 2.8	25.6 ± 3.3	1.023 (0.328)	12	24.6 ± 4.8	25.4 ± 4.6	−0.940 (0.367)
		Female	32	25.2 ± 4.3	26.6 ± 3.5	−4.319 (< 0.001)	33	26.8 ± 3.1	26.5 ± 3.1	0.713 (0.481)	33	26.3 ± 3.5	26.2 ± 4.0	0.180 (0.858)
	Age	<75 years	20	25.2 ± 3.9	27.1 ± 3.0	−4.498 (< 0.001)	16	27.2 ± 2.7	26.6 ± 2.9	1.098 (0.289)	19	25.4 ± 4.3	26.0 ± 4.8	−1.207 (0.243)
		≥75 years	25	25.4 ± 4.3	26.3 ± 3.6	−2.402 (0.024)	29	26.3 ± 3.2	26.1 ± 3.2	0.519 (0.608)	26	26.1 ± 3.7	26.0 ± 3.6	0.228 (0.821)
	Years of Education	≤9 years	26	24.3 ± 4.8	25.8 ± 3.8	−3.545 (0.002)	31	26.4 ± 3.2	26.3 ± 3.3	0.456 (0.651)	30	25.6 ± 3.7	25.6 ± 4.1	−0.11 (0.911)
		>9 years	19	26.7 ± 2.3	27.8 ± 1.9	−3.162 (0.005)	14	27.0 ± 2.5	26.4 ± 2.6	1.979 (0.069)	15	26.3 ± 4.4	26.7 ± 4.3	−0.56 (0.587)
SMCQ	Gender	Male	13	6.1 ± 3.2	4.3 ± 3.2	1.998 (0.069)	12	6.4 ± 2.2	3.7 ± 3.1	4.371 (0.001)	12	5.4 ± 2.2	5.8 ± 2.6	−0.731 (0.480)
		Female	32	5.8 ± 3.4	4.8 ± 3.7	2.025 (0.052)	33	8.0 ± 1.7	5.4 ± 3.2	5.251 (< 0.001)	33	6.7 ± 2.4	6.8 ± 3.2	−0.507 (0.616)
	Age	<75 years	20	5.8 ± 3.0	4.6 ± 3.3	1.842 (0.081)	16	8.1 ± 1.9	5.3 ± 3.3	3.873 (0.002)	19	5.9 ± 2.4	6.3 ± 2.8	−0.812 (0.427)
		≥75 years	25	6.0 ± 3.6	4.8 ± 3.7	2.103 (0.046)	29	7.3 ± 2.0	4.8 ± 3.2	5.358 (< 0.001)	26	6.6 ± 2.4	6.7 ± 3.2	−0.36 (0.722)
	Years of Education	≤9 years	26	6.3 ± 3.5	4.8 ± 3.7	2.562 (0.017)	31	7.4 ± 1.5	4.4 ± 2.9	6.059 (< 0.001)	30	6.4 ± 2.4	6.4 ± 3.1	0.126 (0.901)
		>9 years	19	5.3 ± 3.0	4.5 ± 3.4	1.285 (0.215)	14	8.0 ± 2.8	6.3 ± 3.7	3.067 (0.009)	15	6.1 ± 2.6	6.9 ± 3.0	−1.26 (0.228)
CERAD-K	Gender	Male	13	66.5 ± 17.7	70.0 ± 19.2	−1.963 (0.073)	12	57.7 ± 15.7	63.8 ± 19.8	−3.163 (0.009)	12	62.8 ± 15.3	64.6 ± 16.4	−1.325 (0.212)
		Female	32	68.3 ± 14.2	72.3 ± 12.6	−4.221 (< 0.001)	33	67.0 ± 15.6	70.9 ± 16.4	−4.381 (< 0.001)	33	65.7 ± 13.3	66.5 ± 15.0	−0.951 (0.349)
	Age	<75 years	20	72.1 ± 15.0	74.8 ± 14.2	−2.950 (0.008)	16	65.9 ± 13.3	70.3 ± 13.5	−4.040 (0.001)	19	67.5 ± 15.1	69.0 ± 16.0	−1.661 (0.114)
		≥75 years	25	64.3 ± 14.6	69.1 ± 14.7	−3.688 (0.001)	29	63.7 ± 17.4	68.3 ± 19.5	−3.956 (< 0.001)	26	63.0 ± 12.6	63.8 ± 14.5	−0.69 (0.495)
	Years of Education	≤9 years	26	64.0 ± 14.8	68.8 ± 14.6	−4.206 (< 0.001)	31	65.6 ± 15.7	69.6 ± 17.7	−4.009 (< 0.001)	30	63.5 ± 13.0	64.7 ± 14.9	−1.29 (0.206)
		>9 years	19	72.9 ± 14.2	75.4 ± 14.0	−2.150 (0.045)	14	62.0 ± 17.0	67.6 ± 17.5	−3.657 (0.003)	15	67.7 ± 15.2	68.6 ± 16.1	−0.71 (0.487)
GDSSF-K	Gender	Male	13	3.7 ± 4.3	1.9 ± 2.7	2.599 (0.023)	12	3.0 ± 4.0	3.5 ± 4.8	−0.944 (0.365)	12	4.9 ± 3.6	5.9 ± 4.3	−1.436 (0.179)
		Female	32	4.6 ± 5.0	3.4 ± 3.9	2.121 (0.042)	33	4.2 ± 3.7	5.1 ± 4.9	−1.745 (0.091)	33	4.9 ± 4.1	4.3 ± 4.0	1.187 (0.244)
	Age	<75 years	20	3.8 ± 4.8	3.1 ± 3.0	0.928 (0.365)	16	4.6 ± 4.1	4.4 ± 4.8	0.332 (0.744)	19	6.0 ± 4.6	5.8 ± 4.7	0.203 (0.841)
		≥75 years	25	4.8 ± 4.8	2.8 ± 4.0	3.488 (0.002)	29	3.4 ± 3.5	4.8 ± 5.0	−2.522 (0.018)	26	4.2 ± 3.2	3.9 ± 3.5	0.433 (0.669)
	Years of Education	≤9 years	26	4.7 ± 5.3	2.8 ± 3.7	2.794 (0.010)	31	3.7 ± 3.7	4.5 ± 4.8	−1.511 (0.141)	30	4.9 ± 4.0	4.7 ± 4.3	0.261 (0.796)
		>9 years	19	3.8 ± 4.1	3.1 ± 3.5	1.351 (0.193)	14	4.1 ± 4.1	4.9 ± 5.1	−1.295 (0.218)	15	5.1 ± 3.9	4.7 ± 3.9	0.378 (0.711)

MMSE-DS, Mini-Mental State Examination-Dementia Screening; SMCQ, Subjective Memory Complaint Questionnaire; GDSSF-K, the Geriatric Depression Scale Short Form: Korean Version; CERAD-K, the Korean version of the Consortium to Establish a Registry for Alzheimer’s Disease.

## Data Availability

Data sharing not applicable.

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
