# Peer review of "The Humanoid Robot Sil-Bot in a Cognitive Training Program for Community-Dwelling Elderly People with Mild Cognitive Impairment during the COVID-19 Pandemic: A Randomized Controlled Trial"

_ijerph, 2021, doi:10.3390/ijerph18158198_

Round 1
Reviewer 1 Report
One thing that is unanimously accepted worldwide is that technology has a major impact on medical therapies. This is not only visible in the case of the possibility of developing highly effective drugs but also in the case of the involvement of robots in personalized therapies.
The paper addresses a very interesting aspect, namely the use of robots to improve cognitive function and treat depression in adults over 60 years.
The design of the study by the authors is correct. It complies with both the principles for calculating the sample size and the inclusion and exclusion criteria in the study groups.
The evaluation of the efficacy of the therapy was made on the basis of validated, well-selected questionnaires. Also the techniques of statistical data analysis were correctly applied.
However, these studies have some limitations. One of these is the small number of cases for a randomized study. It is very possible that the effectiveness of Robot-assisted treatment is significantly influenced by the level of education of patients. In this study there are large differences between the number of cases depending on the level of education, which can influence the expected result.
To increase the accuracy of estimating the results for this type of study and especially in randomized studies, larger samples are needed. This is necessary even if the sample size has been calculated, as mentioned by the authors.
It would be interesting to assess whether the difference in pre-post-intervention scores is influenced by certain factors (for example, certain general characteristics of patients).
However, the article makes a contribution to the literature.
Reviewer 2 Report
The authors have presented an interesting randomized clinical trial comparing two modalities of cognitive intervention versus control. The study is interesting but the manuscript presents some deficiencies that should be corrected before its possible publication. Abstract section The reviewer suggests that the authors add a brief background before defining the objective. In the results subsection, the reviewer suggests to the authors, briefly describe the profile of the sample and express the main results numerically, avoiding expressing only the p-value. Introduction section The reviewer considers the section to be excessively long. For this reason, the reviewer suggests that the authors reduce the amount devoted to Mild Cognitive Impairment, giving more importance to previously published interventions. Material and Methods 2.2. Participants The reviewer advises the authors to add the calculation of the power of the study to the one carried out for the sample size. 2.3.3. Statistical Software The reviewer asks the authors if the statistical analysis was done by intention to treat. If not, the authors should reformulate the statistical analysis. Discussion section The reviewer advises authors to draft a limitations subsection within this section, more detailed than that included in conslusions. Likewise, the reviewer suggests that the authors write shorter conclusions and focus on the main results of the study.Author Response
Please see the attachment.
